# A UK survey of young people's views on condom removal during sex

**Farida Ezzat[1] \*, Graham Hart[2], Geraldine Barrett[1] \***

1 Elizabeth Garrett Anderson Institute of Women's Health, University College London, London, United Kingdom, 2 Institute for Global Health, University College London, London, United Kingdom

* g.barrett@ucl.ac.uk (GB); farida.ezzat.22@ucl.ac.uk (FE)

**Data Availability Statement:** The data underlying the results presented in the study are available from the UCL Research Data Repository which can be accessed here: doi 10.5522/04/26326660.

**Funding:** The author(s) received no specific funding for this work.

## Abstract

### Introduction

Non-consensual condom removal (NCCR) refers to the act of removing a condom during sex without the other person's permission. It poses physical and psychological risks to women's health. Views and attitudes regarding this sexual practice are not well understood in the UK. This study aimed to explore young people's views on the morality and criminality of NCCR and how their views are affected by negative health outcomes, relationship status, and socio-demographic characteristics.

### Methods

A quantitative online survey of people aged 18–25 living in the UK was conducted. The survey consisted of two NCCR scenarios, varied by health outcome and relationship status, followed by questions about the morality and criminality of NCCR and respondents' socio-demographic characteristics. Statistical analysis included Chi-square testing and logistic regression modelling.

### Results

Most of the 1729 respondents considered NCCR to be a violation of consent to sex (97.4%-98.1%), to be wrong (99.3%-99.5%), and to be sexual assault (86.3%-89.2%). Respondents were more likely to support prison time for NCCR if the victim got pregnant (52.1%) (rather than depressed (41.6%)) or was part of a casual hook-up (53.9%) (as opposed to a long-term dating relationship (47.2%)). Respondents who were female or non-heterosexual were more likely to view NCCR as sexual assault and support prison as a penalty for NCCR.

### Conclusion

The majority of young UK adults in this survey considered condom removal during sex without the other person's permission to be a violation of consent, morally wrong, and a form of sexual assault. Support for prison as a penalty was lower. These findings can inform future campaigns on consent in sexual relationships and legislation to provide support for women affected by NCCR.

**Competing interests:** The authors have declared that no competing interests exist.

## Introduction

Young people are the most common condom-using demographic [1] and, within that population, the heterosexual male subset frequently engages in condom use resistance [2]. Condom use resistance tactics are employed to avoid using a condom when a sexual partner wants to use one. Davis et al. [3] identified a wide range of condom use resistance tactics including non-consensual condom removal which has recently emerged in the public and legal spheres as a "sex trend" [4].

Non-consensual condom removal (NCCR), sometimes known as *stealthing*, occurs when a person, the *stealther* or NCCR perpetrator, removes the condom before or during sexual intercourse without the permission of the other person, the *stealthed* or NCCR victim. NCCR can occur in heterosexual and male same-sex encounters. Alexandra Brodsky's seminal paper [5] was the first of its kind to offer insight into NCCR through the eyes of stealthed women while contextualizing this sexual practice as a novel form of sexual assault within the American legal system. Brodsky's paper ushered in a new era of research focused on conditional consent, or consent given based on specific conditions such as condom use.

Surveys of female stealthed partners record NCCR rates ranging from 12% to 32% [3, 6–10]. Several risk factors are associated with NCCR. Stealthers are more likely to have a history of sexual aggression and hostile attitudes towards women [11]. They also score higher on psychopathy, antisocial, and borderline personality scales [10, 12]. Alcohol or drug use is correlated with committing as well as experiencing NCCR [8, 13]. In women, experiencing NCCR is associated with being a sex worker [13] and a history of sexual victimization [3]. NCCR is more common in casual or nonexclusive relationships and women who have five or more sexual partners in a single year are more likely to experience NCCR [14–16].

NCCR poses physical and psychological risks to women's health, and potentially that of their male partners. It increases the risk of negative health outcomes for men and women such as sexually transmitted infections (STIs) and, for women, unplanned pregnancy [6, 17, 18]. In terms of psychological outcomes, stealthed women report low levels of control as sexual beings [14] and feelings of sexual shame and violation [19, 20].

Only three studies, employing mixed methods and quantitative methodologies, examined views on NCCR [6, 13, 21]. Evidence from outside the UK shows that many people believe NCCR is wrong due to lack of consent but other prevalent reasons include negative outcomes like STI transmission and unplanned pregnancy [6]. While studies show that most people believe NCCR should be considered sexual assault [13, 21], such views are more common among women than men [13] and less common among NCCR victims than those who have not experienced NCCR [6, 13].

Employing a qualitative approach, Czechowski et al. [6] found that the majority of Canadian undergraduate students thought there should be consequences for stealthers; some mentioned that consequences should apply only if there are negative outcomes like STIs or unplanned pregnancy or if the stealthed partner wants to pursue action against the stealther. Such views display the public's ambivalence to a blanket rule and inclination towards a flexible system to hold stealthers accountable. Interestingly, studies show that less than 2% of stealthed partners report NCCR to the police [6, 16]. This indicates that views on the criminality of NCCR can be influenced, if not restrained, by personal NCCR experiences, rendering the public's views on NCCR and its criminal status, or lack thereof, complex and multifactorial.

The public's views reflect the way that legal systems approach NCCR. In the United States and Canada, the law adopts a risk-based approach requiring bodily harm (e.g. STI transmission or unplanned pregnancy) to consider NCCR an actual crime [5, 6]. The state of California set precedent by making NCCR illegal under civil law in 2021 [22]. A recently passed

Australian legislation states that intentional misrepresentation of condom use vitiates consent [23]. The UK Sexual Offences Act addresses the use of deception during sexual intercourse but does not specify NCCR as a form of sexual assault or rape [24]. However, recent rape convictions demonstrate that the UK legal system is making strides to punish NCCR as a form of rape. In 2019, a British man was sentenced to 12 years in prison when he was found guilty of rape after engaging in NCCR with a female sex worker [25]. In 2024, a UK court convicted a man of rape, sentencing him to over four years in prison for committing NCCR [26].

Existing research on NCCR mainly covers prevalence rates [3, 6–9] and risk factors [3, 8, 12, 13] with few mostly qualitative and non-European studies [6, 13, 21] investigating views on NCCR and their implications on legal reform. To address this gap and extend previous NCCR research, the aim of this study was to explore young people's views in the UK on the morality and criminality of NCCR.

## Methods

### Methodology

A quantitative approach was adopted to determine the prevailing views on NCCR among young people which can guide future awareness campaigns, support services, and legislation addressing NCCR; a cross-sectional survey was designed and delivered online using Qualtrics. There were two eligibility criteria for survey participation: age between 18–25 and residence in the UK.

### Sampling and recruitment

Convenience sampling was used as a time-efficient recruitment method. Participants were recruited via printed flyers distributed on the UCL campus and through social media channels. A post containing the survey link was shared on personal social media accounts. An Instagram account was set up for the study where the survey link was easily accessible to the public. A paid Instagram advertisement, funded by UCL, was launched to boost the survey reach.

### Survey design

This study explored young people's views on several aspects of NCCR: violation of consent to sex, morality, and criminality. Criminality was further subdivided into criminal status–whether NCCR is a form of sexual assault or not–and penalty–whether NCCR perpetrators should face prison time or not. Views were the dependent variables investigated through two independent variables: post-NCCR negative health outcomes for women and relationship status. The survey questionnaire was divided into two sections: NCCR scenarios and socio-demographic questions.

**NCCR scenarios and randomization.** The first section included two fictional scenarios adapted from Nguyen et al. [27]. Scenario A, or the Outcome scenario, focused on negative health outcomes for women. Scenario A1 described an unplanned pregnancy whereas scenario A2 described depression. Scenario B, or the Relationship status scenario, focused on the nature of the relationship between the male and female partners involved in NCCR. Scenario B1 described a casual hook-up whereas scenario B2 described a long-term dating relationship.

Potentially there were four scenarios (A1, A2, B1, B2) to present to survey respondents, which provided challenges in terms of likely respondent fatigue due to the repetitive nature of the scenarios and, more importantly, a likely set of framing effects whereby the answers to a scenario would potentially be affected by knowledge of, and answers given to, previous scenarios. In order to minimize both respondent fatigue and framing effects, we chose to randomize

**Table 1. NCCR scenarios and their variations.**

**Scenario A: *Post-NCCR negative health outcomes for women***
Scenario A1—Unplanned pregnancy
*John and Kate have been having sex with each other for a while. Neither of them have any other sexual partners. On all the occasions they have had sex so far, they have used condoms. One night while having sex, John removed the condom without asking Kate. After sex, Kate discovered the condom on the floor and realized that John had ejaculated inside her vagina. Now, a few weeks later, Kate has just found out that she is pregnant after taking a pregnancy test.*
Scenario A2—Depression
*John and Kate have been having sex with each other for a while. Neither of them have any other sexual partners. On all the occasions they have had sex so far, they have used condoms. One night while having sex, John removed the condom without asking Kate. After sex, Kate discovered the condom on the floor and realized that John had ejaculated inside her vagina. Kate became very upset and anxious about John's actions. In the next few days, she kept revisiting what happened and became depressed.*

**Scenario B: *Relationship status***
Scenario B1—Casual hook-up
*Looking for a casual hookup, Sam and Alice matched on a dating app. They started texting each other for a few days before finally deciding to meet. Their first date goes well. At the end of the night, they went back to Sam's flat to have sex. Alice handed Sam a condom and he put it on. During sex, when Alice wasn't looking, Sam took off the condom without asking her. After Sam ejaculated inside her vagina, he pulled out and was naked and uncovered on the bed. Alice saw that he didn't have the condom on and realized that Sam had taken off the condom during sex.*
Scenario B2—Long-term dating
*Sam and Alice met a few months ago and started dating. They have been having sex with each other every week. As a couple, they buy condoms regularly and agree to use them when they have sex. The most recent time that Sam and Alice had sex, she handed him a condom and he put it on. During sex, when Alice wasn't looking, Sam took off the condom without asking her. After Sam ejaculated inside her vagina, he pulled out and was naked and uncovered on the bed. Alice saw that he didn't have the condom on and realized that Sam had taken off the condom during sex.*

the scenarios, with participants receiving one variation of each scenario (Table 1). This meant that each respondent would only answer two scenarios, thereby decreasing respondent burden and ensuring a fair test between the two versions of each scenario. Further, there would be no framing effects on Scenario A and only one potential framing effect on Scenario B, i.e. the framing effect of Scenario A on Scenario B.

Each scenario was followed by four multiple choice questions:

1. Did X have Y's agreement to sex after he removed the condom without asking her?

2. Did X do something wrong when he removed the condom without asking Y?

3. Did X sexually assault Y when he removed the condom without asking her?

4. Should X serve time in prison for removing the condom without asking Y?

Question 1 addressed violation of consent to sex. To avoid using language that could lead respondents, the words 'consent' and 'consensual' were not included in any part of the survey. Instead, the phrase 'without asking' was used in scenarios and questions as recommended by Czechowski et al. [6]. Question 2 addressed the morality of NCCR while questions 3 and 4 focused on the criminality of NCCR (Fig 1).

Respondents could choose 'Yes', 'No', or 'Not sure' for all questions. Due to the sensitive, sexual nature of the scenarios, all questions were voluntary and respondents could skip any questions they did not wish to answer and still be able to continue the survey.

**Socio-demographic questions.** The second section covered several socio-demographic questions which were taken directly from the UK census [28]. The final question asked whether the respondent was born in the UK or not. All socio-demographic questions were voluntary.

**Participant consent and support.** Before starting the survey, participants were required to check four consent tick boxes which were used to obtain informed consent in an

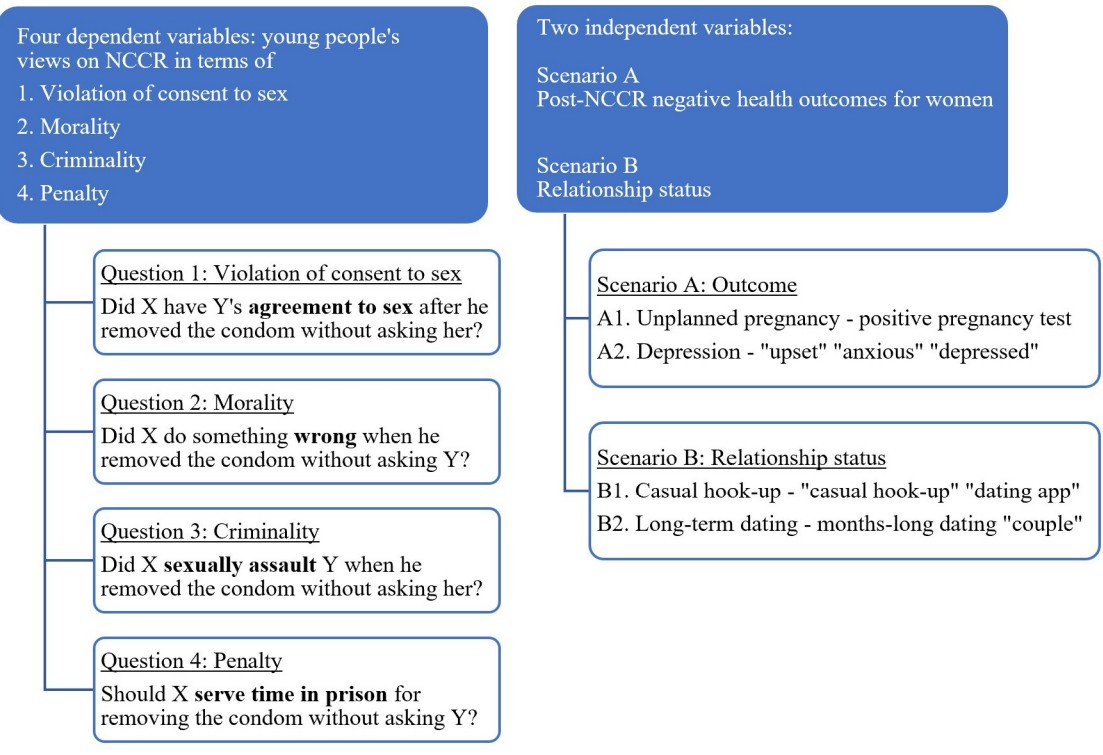

**Fig 1. Operationalization of study variables.**

anonymous manner. The tick boxes ensured that participants met the eligibility criteria, understood the sexual nature of the questions, and knew their responses would be anonymous. Upon completing the survey, participants were shown a brief message thanking them for participating and signposting them to the study's Further Help webpage which was designed to support participants who may find the scenarios emotionally distressing. The webpage included UK support services covering unplanned pregnancy, STIs, and sexual assault.

## Pre-testing

Prior to the online launch of the survey, a pre-testing process was undertaken to optimize question clarity and comprehension. Six pre-testing interviewees were recruited on the UCL campus on 24 and 25 May 2023. All interviewees found the survey easy to understand and straightforward. Survey comprehension was satisfactory and no changes were made to the questions.

## Data collection

The survey was conducted from 30 May to 4 July 2023, over a five-week period. Collected data was stored on a password-protected Qualtrics account accessible to the researchers only. After the survey was closed, the data were exported for analysis.

## Data analysis

Survey responses with at least one answered question were included in the data analysis. Some respondents typed ages outside the target range; these responses were classified as 'Age

unclear' but were not excluded as all respondents checked the consent tick box confirming they are aged 18–25, thus their responses were attributed to possible typing errors.

Chi-square tests were used to identify significant associations between scenario variations and participant views. Where the Outcome scenario was found to significantly affect views, a stratified Chi-square test was used to explore whether there was a potential framing effect of the Outcome scenario on the corresponding view in the Relationship status scenario, which was then investigated formally in later modelling.

Using STATA, unadjusted odds ratios of the dependent variables by socio-demographic characteristics were calculated and logistic regression modelling was performed to calculate adjusted odds ratios with 95% confidence intervals. Adjusted proportions with 95% confidence intervals are also presented. Due to the overwhelming agreement on questions 1 and 2, logistic regression modelling focused on questions 3 and 4 where meaningful associations could be identified. The dependent variable was coded 1 (yes) and 0 (no and no sure).

As a potential framing effect of the Outcome scenario on the penalty question in the Relationship status scenario had been identified, it was explored formally with an interaction term in the model (S1 Table). Informed by this non-significant finding, we used a composite variable of received scenario combinations, i.e. A1+B1, A1+B2, etc. in the final model (Table 8).

## Ethics approval

The study was approved by the UCL Research Ethics Committee, ethics project ID: 24257/002. All participants in the study (pre-testing interviewees and survey participants) were fully informed of the aims and conditions of the study via Participant Information Sheets. The pre-testing interviewees completed and signed written consent forms in the presence of FE. Online survey participants were anonymous. As approved by the research ethics committee, survey participants were required to check four statements in order to continue to the survey questions. The four statements were: "I am aged 18–25", "I live in the UK", "I understand there will be questions about examples of sexual behaviour and condom use", and "I understand that my answers will be anonymous". The answering of survey questions implied consent, as is usual practice in the UK and as was approved by the research ethics committee.

## Results

### Sample

There were 1818 survey responses to the online survey. Of those, 89 respondents checked the consent tick boxes without answering any survey questions; these responses were excluded from data analysis. The remaining 1729 responses were included in data analysis as they contained at least one answered question. The sample was predominantly female, cisgender, non-heterosexual, White, non-religious, and born in the UK (Table 2).

**Scenario completion.** The Outcome scenario was completed by 1729 respondents (S1 Fig), 50.3% answered the Pregnancy variation (scenario A1) and 49.7% answered the Depression variation (scenario A2). The Relationship status scenario was completed by 1693 respondents, 49.9% answered the Casual hook-up variation (scenario B1) and 50.1% answered the Long-term dating variation (scenario B2). No significant associations between scenario allocation and socio-demographic characteristics (S2 Table) were found.

### Survey findings

**Outcome scenario.** The majority of respondents agreed that NCCR is a violation of consent to sex, morally wrong, and a form of sexual assault (Table 3). No significant difference in

**Table 2. Respondents' socio-demographic characteristics.**

| Socio-demographic characteristic | n | % |
|---|---|---|
| **Age** (n = 1667) | | |
| 18 | 140 | 8.4 |
| 19 | 155 | 9.3 |
| 20 | 202 | 12.1 |
| 21 | 208 | 12.5 |
| 22 | 209 | 12.5 |
| 23 | 221 | 13.3 |
| 24 | 289 | 17.3 |
| 25 | 232 | 13.9 |
| Unclear | 11 | 0.8 |
| **Sex** (n = 1668) | | |
| Male | 402 | 24.1 |
| Female | 1266 | 75.9 |
| **Gender identity** (n = 1669) | | |
| Cisgender | 1497 | 89.7 |
| Other gender identity | 172 | 10.3 |
| **Sexual orientation** (n = 1669) | | |
| Straight/heterosexual | 755 | 45.2 |
| Gay or lesbian | 131 | 7.8 |
| Bisexual | 664 | 39.8 |
| Other sexual orientation | 119 | 7.1 |
| **Ethnicity** (n = 1667) | | |
| White | 1416 | 84.9 |
| Mixed or multiple ethnic groups | 84 | 5.0 |
| Asian or Asian British | 87 | 5.2 |
| Black, Black British, Caribbean or African | 22 | 1.3 |
| Other ethnic group | 27 | 1.6 |
| Prefer not to say | 31 | 1.9 |
| **Belonging to a religion** (n = 1666) | | |
| Yes | 286 | 17.2 |
| No | 1380 | 82.8 |
| **Current student** (n = 1665) | | |
| Yes | 917 | 55.1 |
| No | 748 | 44.9 |
| **Current programme of study** (n = 912) | | |
| GCSE/O level/National 5 | 2 | 0.2 |
| College or further education qualification | 37 | 4.1 |
| A/AS/S levels/International Baccalaureate | 64 | 7.0 |
| Undergraduate degree or nursing or teaching qualification | 621 | 68.1 |
| Postgraduate degree | 173 | 19.0 |
| Other | 15 | 1.6 |
| **Highest educational qualification** (n = 1656) | | |
| GCSE/O level/National 5 | 95 | 5.7 |
| College or further education qualification | 159 | 9.6 |
| A/AS/S levels/International Baccalaureate | 615 | 37.2 |
| Undergraduate degree or nursing or teaching qualification | 549 | 33.2 |
| Postgraduate degree | 220 | 13.3 |
| Other | 18 | 1.1 |

(*Continued*)

**Table 2.** (Continued)

| Socio-demographic characteristic | n | % |
|---|---|---|
| **Born in the UK** (n = 1656) | | |
| Yes | 1375 | 83.0 |
| No | 281 | 17.0 |

responses between scenario variations were found in these three questions. Regarding prison as a penalty, there was a significant difference by scenario: over half of the respondents answered 'Yes' in the Pregnancy scenario whereas over half of the respondents answered 'Not sure' or 'No' in the Depression scenario (Table 3).

Respondents who were female, non-heterosexual (i.e. gay, lesbian, bisexual, etc.), or born in the UK were more likely to believe NCCR is sexual assault (Table 4).

Female respondents were more likely to support prison as a penalty for NCCR (Table 5). The Outcome scenario had a statistically significant (p-value <0.001) effect on respondents' views on prison as a penalty; respondents were less likely to support prison time for NCCR in cases where the outcome was depression as opposed to pregnancy (Table 5).

**Relationship status scenario.** The majority of respondents agreed that NCCR is a violation of consent to sex, morally wrong, and a form of sexual assault (Table 6). No significant difference in responses between scenario variations were found in these three questions. Regarding prison as a penalty, there was a significant difference by scenario: over half of the respondents answered 'Yes' in the Casual hook-up scenario, whereas over half of the respondents answered 'Not sure' or 'No' in the Long-term dating scenario (Table 6).

Respondents who were female, non-heterosexual, or born in the UK were more likely to believe NCCR is sexual assault (Table 7). Respondents who were female or non-heterosexual were more likely to support prison time for NCCR (Table 8).

**Table 3. Respondents' views in the outcome scenario.**

| | Yes | | Not sure | | No | |
|---|---|---|---|---|---|---|
| | n | % | n | % | n | % |
| *Did John have Kate's agreement to sex after he removed the condom without asking her?* | | | | | | |
| **Pregnancy** (n = 869) | 8 | 0.9 | 10 | 1.2 | 851 | 97.9 |
| **Depression** (n = 860) | 7 | 0.8 | 9 | 1.0 | 844 | 98.1 |
| | | | | | | Chi = 0.101, df = 2, p-value = 0.95 |
| *Did John do something wrong when he removed the condom without asking Kate?* | | | | | | |
| **Pregnancy** (n = 869) | 864 | 99.4 | 1 | 0.1 | 4 | 0.5 |
| **Depression** (n = 860) | 854 | 99.3 | 3 | 0.3 | 3 | 0.3 |
| | | | | | | Chi = 0.154, df = 2, p-value = 0.56 |
| *Did John sexually assault Kate when he removed the condom without asking her?* | | | | | | |
| **Pregnancy** (n = 869) | 775 | 89.2 | 65 | 7.5 | 29 | 3.3 |
| **Depression** (n = 860) | 742 | 86.3 | 92 | 10.7 | 26 | 3.0 |
| | | | | | | Chi = 5.478, df = 2, p-value = 0.07 |
| *Should John serve time in prison for removing the condom without asking Kate?* | | | | | | |
| **Pregnancy** (n = 869) | 453 | 52.1 | 317 | 36.5 | 99 | 11.4 |
| **Depression** (n = 860) | 358 | 41.6 | 385 | 44.8 | 117 | 13.6 |
| | | | | | | Chi = 19.169, df = 2, p-value <0.001 |

**Table 4. Factors associated with viewing NCCR as sexual assault in the outcome scenario.**

| Variable | Unadjusted | | Adjusted | | Adjusted Proportion |
|---|---|---|---|---|---|
| | OR (95% CI) | P-value | OR (95% CI) | P-value | % (95% CI) |
| **Outcome** | | | | | |
| Pregnancy | 1.0 | 0.07 | 1.0 | 0.241 | 90.6% (88.4–92.4%) |
| Depression | 0.76 (0.57–1.02) | | 0.83 (0.61–1.13) | | 88.9% (86.5–90.9%) |
| **Sex** | | | | | |
| Male | 1.0 | <0.001 | 1.0 | <0.001 | 83.0% (78.9–86.5%) |
| Female | 2.53 (1.85–3.44) | | 2.18 (1.57–3.04) | | 91.4% (89.7–92.9%) |
| **Gender identity** | | | | | |
| Cisgender | 1.0 | 0.02 | 1.0 | 0.26 | 89.7% (87.9–91.2%) |
| Other gender identity | 2.06 (1.10–3.87) | | 1.51 (0.74–3.09) | | 91.2% (84.5–95.1%) |
| **Sexual orientation** | | | | | |
| Heterosexual | 1.0 | <0.001 | 1.0 | <0.001 | 85.3% (82.3–87.8%) |
| Gay or lesbian | 1.74 (0.97–3.12) | | 1.90 (1.03–3.50) | | 91.6% (86.0–95.2%) |
| Bisexual | 3.52 (2.41–5.16) | | 2.84 (1.91–4.22) | | 94.3% (92.1–95.9%) |
| Other sexual orientation | 1.44 (0.81–2.56) | | 1.00 (0.53–1.92) | | 85.3% (76.3–91.2%) |
| **Ethnicity** | | | | | |
| White | 1.0 | 0.03 | 1.0 | 0.95 | 89.9% (88.1–91.4%) |
| Non-White | 0.66 (0.45–0.96) | | 0.99 (0.65–1.51) | | 89.5% (85.2–92.7%) |
| **Belonging to a religion** | | | | | |
| Yes | 1.0 | 0.20 | - | - | - |
| No | 1.28 (0.88–1.85) | | - | - | - |
| **Current student** | | | | | |
| Yes | 1.0 | 0.07 | - | - | - |
| No | 1.33 (0.98–1.81) | | - | - | - |
| **Born in the UK** | | | | | |
| Yes | 1.0 | <0.001 | 1.0 | <0.001 | 90.1% (89.2–92.4%) |
| No | 0.48 (0.34–0.67) | | 0.48 (0.33–0.70) | | 82.8% (77.6–87.0%) |

When used to detect possible framing effects of the Outcome scenario, the Chi-square test yielded a value of 0.005 indicating a significant association between the received Outcome scenario and penalty views in the subsequent Relationship status scenario (data not shown). The composite variable, representing the four possible scenario combinations, remained significant after adjustment (Table 8). Receiving the prior pregnancy scenario and the casual hook-up relationship scenario both increased the likelihood of supporting prison as a penalty; scenario combinations appeared to work in an additive fashion (Table 8).

**Table 5. Factors associated with agreeing that prison should be the penalty for NCCR in the outcome scenario.**

| Variable | Unadjusted | | Adjusted | | Adjusted Proportion |
|---|---|---|---|---|---|
| | OR (95% CI) | P-value | OR (95% CI) | P-value | % (95% CI) |
| **Outcome** | | | | | |
| Pregnancy | 1.0 | <0.001 | 1.0 | <0.001 | 51.7% (48.3–55.1%) |
| Depression | 0.65 (0.54–0.79) | | 0.70 (0.57–0.85) | | 42.8% (39.4–46.3%) |
| **Sex** | | | | | |
| Male | 1.0 | <0.001 | 1.0 | <0.001 | 36.4% (31.8–41.4%) |
| Female | 1.93 (1.53–2.43) | | 1.78 (1.40–2.26) | | 50.8% (48.1–53.6%) |
| **Gender identity** | | | | | |
| Cisgender | 1.0 | 0.87 | - | - | - |
| Other gender identity | 1.15 (0.84–1.58) | | - | | - |
| **Sexual orientation** | | | | | |
| Heterosexual | 1.0 | 0.001 | 1.0 | 0.07 | 44.1% (40.5–47.7%) |
| Gay or lesbian | 0.92 (0.63–1.34) | | 1.00 (0.68–1.48) | | 44.2% (35.7–53.1%) |
| Bisexual | 1.45 (1.18–1.79) | | 1.31 (1.06–1.62) | | 50.8% (46.9–54.7%) |
| Other sexual orientation | 1.53 (1.04–2.26) | | 1.35 (0.91–2.01) | | 51.6% (42.6–60.6%) |
| **Ethnicity** | | | | | |
| White | 1.0 | 0.99 | - | - | - |
| Non-White | 1.00 (0.76–1.31) | | - | | - |
| **Belonging to a religion** | | | | | |
| Yes | 1.0 | 0.86 | - | - | - |
| No | 0.98 (0.76–1.26) | | - | - | |
| **Current student** | | | | | |
| Yes | 1.0 | 0.22 | - | - | - |
| No | 0.89 (0.73–1.07) | | - | - | |
| **Born in the UK** | | | | | |
| Yes | 1.0 | 0.17 | - | - | - |
| No | 0.84 (0.65–1.08) | | - | - | |

## Discussion

The majority of respondents agreed that NCCR is a violation of consent to sex, morally wrong, and criminal. Respondents who were female, non- heterosexual, or were born in the UK were more likely to view NCCR as sexual assault. Outcome and relationship status significantly affected penalty views; pregnancy and casual hook-up relationships increased the likelihood of supporting prison as a penalty for NCCR. Respondents who were female or non-heterosexual were more likely to support prison as a penalty for NCCR. A framing effect of the Outcome scenario was established in relation to penalty views in the subsequent Relationship status scenario.

## Consent: It's complicated

Regardless of outcome or relationship status, the overwhelming majority of respondents believed that NCCR is a violation of consent to sex and morally wrong. This finding is

**Table 6. Respondents' views in the relationship status scenario.**

| | Yes | | Not sure | | No | |
|---|---|---|---|---|---|---|
| | **n** | **%** | **n** | **%** | **n** | **%** |
| | *Did Sam have Alice's agreement to sex after he removed the condom without asking her?* | | | | | |
| **Casual hook-up** (n = 844) | 11 | 1.3 | 6 | 0.7 | 827 | 98.0 |
| **Long-term dating** (n = 849) | 16 | 1.9 | 6 | 0.7 | 827 | 97.4 |
| | Chi = 0.911, df = 2, p-value = 0.63 | | | | | |
| | *Did Sam do something wrong when he removed the condom without asking Alice?* | | | | | |
| **Casual hook-up** (n = 844) | 840 | 99.5 | 1 | 0.1 | 3 | 0.4 |
| **Long-term dating** (n = 849) | 845 | 99.5 | 0 | 0 | 4 | 0.5 |
| | Chi = 1.143, df = 2, p-value = 0.57 | | | | | |
| | *Did Sam sexually assault Alice when he removed the condom without asking her?* | | | | | |
| **Casual hook-up** (n = 844) | 753 | 89.2 | 70 | 8.3 | 21 | 2.5 |
| **Long-term dating** (n = 849) | 736 | 86.7 | 80 | 9.4 | 33 | 3.9 |
| | Chi = 3.513, df = 2, p-value = 0.17 | | | | | |
| | *Should Sam serve time in prison for removing the condom without asking Alice?* | | | | | |
| **Casual hook-up** (n = 844) | 455 | 53.9 | 308 | 36.5 | 81 | 9.6 |
| **Long-term dating** (n = 849) | 401 | 47.2 | 335 | 39.5 | 113 | 13.3 |
| | Chi = 9.804, df = 2, p-value = 0.007 | | | | | |

consistent with Czechowski et al. [6] where almost all survey respondents believed NCCR is wrong, over half of whom cited lack of consent as the reason for their view.

Aligning with previous research [21], most respondents in our study viewed NCCR as sexual assault. Although almost all of our respondents confirmed that the male partner no longer had the female partner's agreement to sex after NCCR, not as many viewed it as sexual assault. The gap between identifying violation of consent to sex and viewing NCCR as sexual assault could reflect a mismatch in young people's understanding of consent, which has been previously observed [21].

Several UK-based efforts, such as the #Consentiseverything campaign [29], focus on explicit consent as the main criterion for sexual literacy. This focus is informed by maladaptive consent norms, including interpreting the lack of refusal as consent, which are commonly held by men and young people in the UK [30]. While obtaining verbal consent is deemed important, UK university students often find it awkward to openly discuss consent, preferring to negotiate it using non-verbal cues [31]. However, non-verbal consent can be unclear, and ambiguous consent has been linked to higher odds of sexual assault victimization among undergraduate students [32] while the use of verbal cues has been associated with greater feelings of sexual consent and safety among couples in committed relationships [33]. It is worth noting that communicating consent may not be the ultimate tactic to prevent NCCR. Bogen and Lorenz [9] showed that a female partner's sexual communication skill did not impact perpetration of NCCR or its negative health outcomes; they argue that since NCCR is an intentional "non-communication" act perpetrated by a male partner in defiance of the female partner's conditional consent, campaigns focusing on improving women's consent communication are not appropriate to curb NCCR rates. Therefore, current consent-centric campaigns are misaligned with the reality of sexual dynamics among young people and better sexual health awareness focused on context-specific and conditional consent is needed.

**Table 7. Factors associated with viewing NCCR as sexual assault in the relationship status scenario.**

| Variable | Unadjusted | | Adjusted | | Adjusted Proportion |
|---|---|---|---|---|---|
| | OR (95% CI) | P-value | OR (95% CI) | P-value | % (95% CI) |
| **Relationship status** | | | | | |
| Casual hook-up | 1.0 | 0.11 | 1.0 | 0.24 | 90.5% (88.3–92.4%) |
| Long-term dating | 0.79 (0.59–1.06) | | 0.83 (0.61–1.13) | | 88.7% (86.3–90.7%) |
| **Sex** | | | | | |
| Male | 1.0 | <0.001 | 1.0 | <0.001 | 82.4% (78.3–85.9%) |
| Female | 2.56 (1.88–3.49) | | 2.26 (1.63–3.13) | | 91.3% (89.6–92.8%) |
| **Gender identity** | | | | | |
| Cisgender | 1.0 | 0.02 | 1.0 | 0.23 | 89.5% (97.7–91.1%) |
| Other gender identity | 2.08 (1.11–3.90) | | 1.54 (0.76–3.15) | | 91.2% (84.4–95.1%) |
| **Sexual orientation** | | | | | |
| Heterosexual | 1.0 | <0.001 | 1.0 | <0.001 | 85.1% (82.1–87.6%) |
| Gay or lesbian | 1.75 (0.98–3.15) | | 1.91 (1.04–3.52) | | 91.6% (85.9–95.1%) |
| Bisexual | 3.66 (2.49–5.38) | | 1.96 (1.96–4.36) | | 94.3% (92.2–95.6%) |
| Other sexual orientation | 1.35 (0.77–2.36) | | 0.93 (0.49–1.74) | | 84.1% (74.9–90.3%) |
| **Ethnicity** | | | | | |
| White | 1.0 | 0.05 | - | | - |
| Non-White | 0.69 (0.47–1.00) | | - | | - |
| **Belonging to a religion** | | | | | |
| Yes | 1.0 | 0.81 | - | - | - |
| No | 1.05 (0.71–1.55) | | - | - | |
| **Current student** | | | | | |
| Yes | 1.0 | 0.40 | - | - | - |
| No | 1.14 (0.84–1.54) | | - | - | |
| **Born in the UK** | | | | | |
| Yes | 1.0 | 0.001 | 1.0 | 0.01 | 90.5% (88.8–92.0%) |
| No | 0.56 (0.40–0.80) | | 0.58 (0.40–0.83) | | 84.5% (79.8–88.3%) |

## Constructing criminality

In agreement with previous research [6, 21], female respondents in our study were more likely to view NCCR as sexual assault. One possible explanation for this finding may be that more women experience NCCR compared to men and are in turn more negatively affected by it [6, 7, 13]. Similarly, respondents who identified as gay, lesbian, or bisexual were more likely to view NCCR as sexual assault compared to their heterosexual counterparts. Such increased awareness could be due to the fact that, not unlike women, sexual minorities are disproportionately affected by sexual assault [34–36].

**Table 8. Factors associated with agreeing that prison should be the penalty for NCCR in the relationship scenario.**

| Variable | Unadjusted | | Adjusted | | Adjusted Proportion |
|---|---|---|---|---|---|
| | OR (95% CI) | P-value | OR (95% CI) | P-value | % (95% CI) |
| **Scenarios Received*** | | | | | |
| A1-B1 | 1.0 | 0.0001 | 1.0 | 0. 0005 | 57.9% (53.0–62.7%) |
| A1-B2 | 0.75 (0.58–0.99) | | 0.76 (0.58–1.00) | | 51.3% (46.5–56.0%) |
| A2-B1 | 0.69 (0.52–0.90) | | 0.72 (0.55–0.95) | | 50.0% (45.1–54.8%) |
| A2-B2 | 0.53 (0.40–0.69) | | 0.55 (0.41–0.73) | | 43.1% (38.4–48.0%) |
| **Sex** | | | | | |
| Male | 1.0 | <0.001 | 1.0 | <0.001 | 41.1% (36.3–46.1%) |
| Female | 1.78 (1.41–2.23) | | 1.62 (1.28–2.06) | | 53.6% (50.8–56.4%) |
| **Gender identity** | | | | | |
| Cisgender | 1.0 | 0.34 | - | - | - |
| Other gender identity | 1.17 (0.85–1.60) | | - | | - |
| **Sexual orientation** | | | | | |
| Heterosexual | 1.0 | <0.001 | 1.0 | 0.005 | 46.1% (42.5–49.8%) |
| Gay or lesbian | 1.00 (0.69–1.45) | | 1.05 (0.71–1.53) | | 47.2% (38.6–56.1%) |
| Bisexual | 1.61 (1.31–1.99) | | 1.46 (1.17–1.81) | | 55.5% (51.6–59.3%) |
| Other sexual orientation | 1.63 (1.10–2.40) | | 1.43 (0.96–2.13) | | 55.1% (45.9–64.0%) |
| **Ethnicity** | | | | | |
| White | 1.0 | 0.87 | - | - | - |
| Non-White | 1.02 (0.78–1.34) | | - | | - |
| **Belonging to a religion** | | | | | |
| Yes | 1.0 | 0.75 | - | - | - |
| No | 0.96 (0.74–1.24) | | - | - | - |
| **Current student** | | | | | |
| Yes | 1.0 | 0.08 | - | - | - |
| No | 0.84 (0.69–1.02) | | - | - | - |
| **Born in the UK** | | | | | |
| Yes | 1.0 | 0.11 | - | - | - |
| No | 0.81 (0.63–1.05) | | - | - | - |

*A1: Pregnancy; A2: Depression; B1: Casual hook-up; B2: Long-term dating

Despite this, experiencing NCCR does not always correspond to acknowledging sexual assault. Research shows that, compared to those who have not experienced NCCR, victims are less likely to view NCCR as sexual assault [13]. Aversion to acknowledge victimization [37], concerns about the negative impact of the sexual assault label [38], and emotional attachment to sexual partners [4, 39] can act as barriers for victims to acknowledge sexual assault.

Respondents born in the UK were more likely to view NCCR as sexual assault which could be attributed to the rise of UK sexual assault awareness campaigns targeting young people. For

example, the 2014 'I Heart Consent' campaign [40] entailed the delivery of consent workshops in 20 UK universities where attendees were able to better discuss sexual consent.

## Consequences and conundrums

This study was the first of its kind to investigate views on prison as a penalty for NCCR. Previous research on mandating consequences for NCCR show a supportive majority [6]. Ambivalent or oppositional views were significantly higher in our study which could be attributed to the specificity and severity of prison time as well as cultural differences between study populations.

Over half of the respondents supported prison time if the female partner got pregnant but that support dropped by almost 10% if she became depressed. This difference shows that pregnancy is perceived as a more serious consequence warranting legal penalty. Conditional support for consequences based on outcome, usually pregnancy, has been previously noted [6]. In Britain, unplanned pregnancy is most common among young people [41] and 40% of women who use emergency contraception cite condom failure as their main reason to do so [42]. This further emphasizes NCCR as a women's health issue. In cases where victims do not discover they have been stealthed, their reproductive health is compromised as they are unable to procure emergency contraception and are therefore at a higher risk of unplanned pregnancy.

Respondents were less likely to support prison if the partners were part of a long-term dating relationship compared to a casual hook-up. This is echoed by some participants in Czechowski et al. [6] whose views on legal consequences for NCCR were influenced by relationship status. Current research shows that NCCR often occurs in casual or short-term relationships [14–16, 20], however the surreptitious nature of NCCR and existing trust between couples may lead to underreporting in stable relationships. Sexual crime scenarios where victim and perpetrator know each other are perceived as less severe, resulting in lower rates of crime reporting and punishment severity [43, 44], thus impeding sexual justice for women in long-term relationships. On the other hand, women outside stable relationships are also at high risk. Women who have multiple sexual partners and those in short-term relationships are more likely to experience both unplanned pregnancy [41] and NCCR [14–16, 20], consequently they are particularly vulnerable to reproductive health problems and sexual assault victimization.

Female respondents were more likely than male respondents to support prison as a penalty. This gendered effect is a novel finding as previous research had not shown differences in such views [6], although female jurors are more likely to convict defendants in sexual assault and rape cases [45, 46]. Non-heterosexual respondents were also more likely to support prison time which aligns with this population's higher tendency to recognize sexual assault as discussed earlier.

## Limitations

This study had several limitations. The use of a non-random convenience sample, dominated by university students and Instagram users, limits the generalizability of the results to the general population of young people aged 18–25. Additionally, the majority of the study population was White and cisgender, further reducing result generalizability to ethnic and gender minorities who are understudied in NCCR research. Nevertheless, this was the first UK study to cover views on NCCR. Its quantitative approach and large sample size offer valuable insight to begin to address research gaps in this area.

## Conclusion

NCCR has been identified as the most frequently reported form of condom nonuse [47], further underlining the urgency of investigating this phenomenon. Since personal NCCR experiences can influence views [13], future studies should focus on NCCR victims and perpetrators to better understand their legal expectations and guide policy reform accordingly [48]. Given the prevailing awareness that NCCR is a violation of consent and a form of sexual assault, future sexual health campaigns and legislation should tackle this phenomenon to provide the needed support for women affected by NCCR.

## Supporting information

**S1 Fig. Scenario allocation.**
(DOCX)

**S1 Table. Interaction term logistic regression model of the association between outcome scenario and views on penalty in the relationship status scenario.**
(DOCX)

**S2 Table. Associations between sociodemographic factors and scenario allocation.**
(DOCX)

## Acknowledgments

This research was conducted as part of the UCL EGA Institute for Women's Health MSc Women's Health programme.

## Author Contributions

**Conceptualization:** Farida Ezzat, Graham Hart, Geraldine Barrett.

**Data curation:** Farida Ezzat, Geraldine Barrett.

**Formal analysis:** Farida Ezzat, Geraldine Barrett.

**Investigation:** Farida Ezzat, Geraldine Barrett.

**Methodology:** Farida Ezzat, Graham Hart, Geraldine Barrett.

**Project administration:** Farida Ezzat, Geraldine Barrett.

**Supervision:** Geraldine Barrett.

**Visualization:** Farida Ezzat, Geraldine Barrett.

**Writing – original draft:** Farida Ezzat.

**Writing – review & editing:** Farida Ezzat, Graham Hart, Geraldine Barrett.

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
