## [Decision Letter · Decision Letter 0]

17 Apr 2024

PONE-D-24-03228A UK survey of young people’s views on condom removal during sexPLOS ONE

Dear Dr. Barrett,

Thank you for submitting your manuscript to PLOS ONE. After careful consideration, we feel that it has merit but does not fully meet PLOS ONE’s publication criteria as it currently stands. Therefore, we invite you to submit a revised version of the manuscript that addresses the points raised during the review process.

In addition to the comments provided by Reviewer #1, to which I ask authors to respond carefully and in detail, I also ask authors to consider the following:

I agree with Reviewer #1 that the Introduction be extremely concise and that the previous research be better explored, perhaps even stating what methods of exploring the products of the literature were adopted (e.g., which Databases or Register were queried and with which text strings, etc.) for stating that “Existing research on NCCR mainly covers prevalence rates and risk factors with few studies, mostly qualitative and North America-specific, investigating views on NCCR and their implications on legal reform.”Moreover, after the statement “Existing research on NCCR mainly covers prevalence rates and risk factors with few studies” report the bibliographical references to the “few studies.” After the statement “mostly qualitative and North America-specific, investigating views on NCCR and their implications on legal reform” report the bibliographical references.I would suggest moving the “Aim and Objectives” session to the end of the Introduction, after the rationale.Also, the distinction between “aim and objectives” is not clear to me. Does “aim” mean general objective? I would suggest rephrasing with the terms “objective” or “aim” or “goal” to mean the general objective and rephrasing the objectives as “predictions” or “expectations”, trying to add to each of them a bibliographic reference that justifies it.In addition to defining the study as having “A quantitative approach,” the type of study, e.g., “cross-sectional,” etc., should also be clarified.Finally, the Discussion should be updated based on the literature products suggested by Reviewer #1 and any new exploration of the literature.

We look forward to receiving your revised manuscript.

Kind regards,

Stefano Federici, Ph.D.

Academic Editor

PLOS ONE

2. Please note that your Data Availability Statement is currently missing [the repository name and/or the DOI/accession number of each dataset OR a direct link to access each database]. If your manuscript is accepted for publication, you will be asked to provide these details on a very short timeline. We therefore suggest that you provide this information now, though we will not hold up the peer review process if you are unable.

Additional Editor Comments:

3. In addition to the comments provided by Reviewer #1, to which I ask authors to respond carefully and in detail, I also ask authors to consider the following:

• I agree with Reviewer #1 that the Introduction be extremely concise and that the previous research be better explored, perhaps even stating what methods of exploring the products of the literature were adopted (e.g., which Databases or Register were queried and with which text strings, etc.) for stating that “Existing research on NCCR mainly covers prevalence rates and risk factors with few studies, mostly qualitative and North America-specific, investigating views on NCCR and their implications on legal reform.”

• Moreover, after the statement “Existing research on NCCR mainly covers prevalence rates and risk factors with few studies” report the bibliographical references to the “few studies.” After the statement “mostly qualitative and North America-specific, investigating views on NCCR and their implications on legal reform” report the bibliographical references.

• I would suggest moving the “Aim and Objectives” session to the end of the Introduction, after the rationale.

• Also, the distinction between “aim and objectives” is not clear to me. Does “aim” mean general objective? I would suggest rephrasing with the terms “objective” or “aim” or “goal” to mean the general objective and rephrasing the objectives as “predictions” or “expectations”, trying to add to each of them a bibliographic reference that justifies it.

• In addition to defining the study as having “A quantitative approach,” the type of study, e.g., “cross-sectional,” etc., should also be clarified.

• Finally, the Discussion should be updated based on the literature products suggested by Reviewer #1 and any new exploration of the literature.

Reviewers' comments:

Reviewer's Responses to Questions

**Comments to the Author**

1. Is the manuscript technically sound, and do the data support the conclusions?

Reviewer #1: Yes

2. Has the statistical analysis been performed appropriately and rigorously? 

Reviewer #1: I Don't Know

3. Have the authors made all data underlying the findings in their manuscript fully available?

Reviewer #1: Yes

4. Is the manuscript presented in an intelligible fashion and written in standard English?

Reviewer #1: Yes

5. Review Comments to the Author

Reviewer #1: Dear authors,

Thank you for conducting this line of research and for using a UK sample! I would very much like to see your research extended and using a sample above 25 years of age.

Overall, very well done! I am recommending minor revisions.

Introduction

Good and well organised but too short. Have you looked at all previous research? This is just a general question as I understand that there is not much research on this area.

Methods

It is not clear to me why you chose to present scenario variations randomised to participants rather than presenting all scenarios to all the participants. Can you please provide an explanation for this? As currently stands, randomisation does not really help your research as you were not looking to infer causality. I understand that you might have chosen to do so due to time limitation reasons, which is absolutely fine! Is that the case?

Results

You mention you conducted chi-square tests and logistic regressions. I have checked the supplementary material, but this is not how regressions should be presented as a lot of information is missing. Also, there is no information or results on the chi-square tests. I am sure that you have conducted the correct calculations, but can you please add the missing information?

Discussion

Your discussion is well structured. Consider adding the following research:

https://doi.org/10.1093/pubmed/fdab361

https://doi.org/10.1177/08862605211044101

https://doi.org/10.1080/0092623X.2021.1937417

https://doi.org/10.1080/13676261.2022.2152317

6. PLOS authors have the option to publish the peer review history of their article (what does this mean?). If published, this will include your full peer review and any attached files.

Reviewer #1: No

---

## [Author Response · Author response to Decision Letter 0]

19 Jul 2024

Reviewer #1: Dear authors,

Thank you for conducting this line of research and for using a UK sample! I would very much like to see your research extended and using a sample above 25 years of age.

Overall, very well done! I am recommending minor revisions.

Our response: Thank you!

Reviewer: Introduction

Good and well organised but too short. Have you looked at all previous research? This is just a general question as I understand that there is not much research on this area.

Our response: In our original manuscript, we had aimed for the length of the introduction to be in keeping with the balance of other PloS ONE papers; indeed we had excluded much information which we could have included. We have now substantially extended our introduction to include a more comprehensive review of NCCR studies as well as recent legal cases.

Reviewer: Methods

It is not clear to me why you chose to present scenario variations randomised to participants rather than presenting all scenarios to all the participants. Can you please provide an explanation for this? As currently stands, randomisation does not really help your research as you were not looking to infer causality. I understand that you might have chosen to do so due to time limitation reasons, which is absolutely fine! Is that the case?

Our response: We have changed the “NCCR Scenarios” section in the methods to “NCCR Scenarios and randomization” and now include a new paragraph explaining the rationale for randomization:

“Potentially there were four scenarios (A1, A2, B1, B2) to present to survey respondents, which provided challenges in terms of likely respondent fatigue due to the repetitive nature of the scenarios and, more importantly, a likely set of framing effects whereby the answers to a scenario would potentially be affected by knowledge of, and answers given to, previous scenarios. In order to minimise both respondent fatigue and framing effects, we chose to randomize the scenarios, with participants receiving one variation of each scenario (Table 1). This meant that each respondent would only answer two scenarios, thereby decreasing respondent burden and ensuring a fair test between the two versions of each scenario. Further, there would be no framing effects on Scenario A and only one potential framing effect on Scenario B, i.e. the framing effect of Scenario A on Scenario B.” 

Reviewer: Results

You mention you conducted chi-square tests and logistic regressions. I have checked the supplementary material, but this is not how regressions should be presented as a lot of information is missing. Also, there is no information or results on the chi-square tests. I am sure that you have conducted the correct calculations, but can you please add the missing information?

Our response: The results of the Chi-square tests are presented in table 3 and table 6. The Chi-square test used to explore the potential framing effect of the Outcome scenario on the corresponding view in the Relationship status scenario is reported in the text relating to the Relationship scenario (Lines 271-273); although the p-value is reported, we do not display substantive data and therefore have added “data not shown”. The framing effect is then explored formally through the logistic regression models (Table S1 and Table 8)

The results of the various logistic regression models are presented in tables 4, 5, 7, 8 and S1. We have presented the logistic regression models as odds ratios with 95% confidence intervals and associated p-values; this has been our standard way of presenting logistic regression models in many published papers, including in PloS ONE. We also presented adjusted proportions in tables 4, 5, 7 and 8 as these may be an easier way for some readers to comprehend the data.

We have made small changes to the text in the Methods’ “data analysis” section, for clarity. We have also amended some of the titles to the tables to improve clarity.

Reviewer: Discussion

Your discussion is well structured. Consider adding the following research:

https://doi.org/10.1093/pubmed/fdab361

https://doi.org/10.1177/08862605211044101

https://doi.org/10.1080/0092623X.2021.1937417

https://doi.org/10.1080/13676261.2022.2152317

Our response: Thank you for the suggested references, all of which touched on themes of sexual consent. Two of the references (Willis et al, 2021 and Willis and Marcantonio, 2021) were directly relevant and within the scope of our paper and so are now included.

Editor: Additional Editor Comments:

3. In addition to the comments provided by Reviewer #1, to which I ask authors to respond carefully and in detail, I also ask authors to consider the following:

• I agree with Reviewer #1 that the Introduction be extremely concise and that the previous research be better explored, perhaps even stating what methods of exploring the products of the literature were adopted (e.g., which Databases or Register were queried and with which text strings, etc.) for stating that “Existing research on NCCR mainly covers prevalence rates and risk factors with few studies, mostly qualitative and North America-specific, investigating views on NCCR and their implications on legal reform.”

Our response: Thank you for the advice. As per our earlier response, we have extended the introduction. The sentence about which more information is requested has now become a paragraph. As we were reporting findings as part of an introduction, rather than presenting the findings of a systematic review, we did not carry out (and therefore did not record) a systematic search, i.e. recording all databases used, all search terms used, with ensuing formal review of abstracts and full texts with subsequent data abstraction. 

Editor • Moreover, after the statement “Existing research on NCCR mainly covers prevalence rates and risk factors with few studies” report the bibliographical references to the “few studies.” After the statement “mostly qualitative and North America-specific, investigating views on NCCR and their implications on legal reform” report the bibliographical references.

Our response: This sentence has now been amended with references added.

Editor: • I would suggest moving the “Aim and Objectives” session to the end of the Introduction, after the rationale.

Our response: The last sentence of the introduction (“To address this gap and extend previous NCCR research, this study explored how young people in the UK view NCCR.”) has now been changed to include the exact study aim (“To address this gap and extend previous NCCR research, the aim of this study was to explore young people’s views in the UK on the morality and criminality of NCCR.”) 

Editor: • Also, the distinction between “aim and objectives” is not clear to me. Does “aim” mean general objective? I would suggest rephrasing with the terms “objective” or “aim” or “goal” to mean the general objective and rephrasing the objectives as “predictions” or “expectations”, trying to add to each of them a bibliographic reference that justifies it.

Our response: The aim an objectives that were presented in our original manuscript were those approved by the UCL Research Ethics Committee and therefore we do not feel able to rewrite them post-hoc. However, the overall aim is adequate for the paper (and is now presented in the introduction, as requested) and we have deleted the objectives.

Editor: • In addition to defining the study as having “A quantitative approach,” the type of study, e.g., “cross-sectional,” etc., should also be clarified.

Our response: In our original manuscript, we had stated the type of study (survey) in the second sentence. We have now joined the second sentence to the first, and we have added the word “cross-sectional”.

Editor: • Finally, the Discussion should be updated based on the literature products suggested by Reviewer #1 and any new exploration of the literature.

Our response: The discussion has been updated following the incorporation of additional studies related to NCCR.

---

## [Decision Letter · Decision Letter 1]

6 Aug 2024

PONE-D-24-03228R1A UK survey of young people’s views on condom removal during sexPLOS ONE

Dear Dr. Barrett,

Thank you for submitting your manuscript to PLOS ONE. After careful consideration, we feel that it has merit but does not fully meet PLOS ONE’s publication criteria as it currently stands. Therefore, we invite you to submit a revised version of the manuscript that addresses the points raised during the review process.

Just a little more effort to make your paper suitable for publication. I invite the authors to consider the reviewer's suggestion regarding the need to expand the introduction by providing the requested clarifications.

We look forward to receiving your revised manuscript.

Kind regards,

Stefano Federici, Ph.D.

Academic Editor

PLOS ONE

Journal Requirements:

Additional Editor Comments:

Just a little more effort to make your paper suitable for publication. I invite the authors to consider the reviewer's suggestion regarding the need to expand the introduction by providing the requested clarifications.

Reviewers' comments:

Reviewer's Responses to Questions

**Comments to the Author**

1. If the authors have adequately addressed your comments raised in a previous round of review and you feel that this manuscript is now acceptable for publication, you may indicate that here to bypass the “Comments to the Author” section, enter your conflict of interest statement in the “Confidential to Editor” section, and submit your "Accept" recommendation.

Reviewer #1: All comments have been addressed

2. Is the manuscript technically sound, and do the data support the conclusions?

Reviewer #1: Yes

3. Has the statistical analysis been performed appropriately and rigorously? 

Reviewer #1: Yes

4. Have the authors made all data underlying the findings in their manuscript fully available?

Reviewer #1: Yes

5. Is the manuscript presented in an intelligible fashion and written in standard English?

Reviewer #1: Yes

6. Review Comments to the Author

Reviewer #1: Dear authors,

Thank you for addressing all my comments. However, there are still issues with the Introduction as it feels to be too short and not in-depth enough. I'd like to see a short description of the methods used in the research you have chosen to include in your Introduction. At the moment it is hard to understand how the previous research has been conducted, i.e. qual or quant etc. There is no need to present the info as a systematic review (because this is not what you did), but you need to present the info in a more systematic way.

Other than that, I am satisfied with the way you have addressed my comments. Thank you!

7. PLOS authors have the option to publish the peer review history of their article (what does this mean?). If published, this will include your full peer review and any attached files.

Reviewer #1: No

---

## [Author Response · Author response to Decision Letter 1]

6 Sep 2024

Reviewer #1: Dear authors,

Thank you for addressing all my comments. However, there are still issues with the Introduction as it feels to be too short and not in-depth enough. I'd like to see a short description of the methods used in the research you have chosen to include in your Introduction. At the moment it is hard to understand how the previous research has been conducted, i.e. qual or quant etc. There is no need to present the info as a systematic review (because this is not what you did), but you need to present the info in a more systematic way.

Other than that, I am satisfied with the way you have addressed my comments. Thank you!

Our response:

We would like to thank the reviewer for their careful reading of our paper.

We have added additional detail of the methodologies of the studies cited in the introduction in two paragraphs of the revised paper. We also note that our statements are referenced, and it is possible for readers to find and read the original research that is being cited if they wish to know more details about the research.

In terms of style and length, our introduction is deliberately in keeping with other recent sexual health-related PloS ONE papers, e.g.: 

https://journals.plos.org/plosone/article?id=10.1371/journal.pone.0259182

https://journals.plos.org/plosone/article?id=10.1371/journal.pone.0282187

https://journals.plos.org/plosone/article?id=10.1371/journal.pone.0189607

---

## [Decision Letter · Decision Letter 2]

17 Sep 2024

A UK survey of young people’s views on condom removal during sex

PONE-D-24-03228R2

Dear Dr. Barrett,

We’re pleased to inform you that your manuscript has been judged scientifically suitable for publication and will be formally accepted for publication once it meets all outstanding technical requirements.

Kind regards,

Stefano Federici, Ph.D.

Academic Editor

PLOS ONE

Additional Editor Comments (optional):

Reviewers' comments:

Reviewer's Responses to Questions

**Comments to the Author**

1. If the authors have adequately addressed your comments raised in a previous round of review and you feel that this manuscript is now acceptable for publication, you may indicate that here to bypass the “Comments to the Author” section, enter your conflict of interest statement in the “Confidential to Editor” section, and submit your "Accept" recommendation.

Reviewer #1: All comments have been addressed

2. Is the manuscript technically sound, and do the data support the conclusions?

Reviewer #1: Yes

3. Has the statistical analysis been performed appropriately and rigorously? 

Reviewer #1: Yes

4. Have the authors made all data underlying the findings in their manuscript fully available?

Reviewer #1: Yes

5. Is the manuscript presented in an intelligible fashion and written in standard English?

Reviewer #1: Yes

6. Review Comments to the Author

Reviewer #1: Dear authors,

You have addressed all my comments and I thank you for that!

Well done!

I hope you will extend your research to a larger age range.

7. PLOS authors have the option to publish the peer review history of their article (what does this mean?). If published, this will include your full peer review and any attached files.

Reviewer #1: No

---

## [Editor Report · Acceptance letter]

23 Sep 2024

PONE-D-24-03228R2 

PLOS ONE

Dear Dr. Barrett, 

I'm pleased to inform you that your manuscript has been deemed suitable for publication in PLOS ONE. Congratulations! Your manuscript is now being handed over to our production team.

Kind regards, 

on behalf of

Prof. Stefano Federici 

Academic Editor

PLOS ONE